# Retention in care among people living with HIV in the national antiretroviral therapy programme in Guinea: A retrospective cohort analysis

Kadio Jean-Jacques Olivier Kadio[1,2], Cissé Amadou[3], Saidou Diallo Thierno[4], Foromo Guilavogui[4], Fapeingou Tounkara Adrien[4], Pe Damey[4], Sow Alhassane[4], Fily Bah Fatoumata[4], Sékou Youla Souleymane[4], Diallo Ibrahima[4], Nestor Leno Niouma[4], Mboungou Lazare[4], Nyawotope Koffi Ahiatsi Arnold[4], Kaba Laye[4], Sy Zeynabou[5,6], Vallès-Casanova Ignasi[7], Wringe Alison[8,9], Hoibak Sarah[9], Koïta Youssouf[1], Xavier Vallès[9,10,11,12]*

1 Guinea Infectious Disease Research and Training Center, Gamal Abdel Nasser University of Conakry, Conakry, Guinea, 2 Department of Public Health and Pharmaceutical Legislation, Faculty of Health Sciences and Technology, Gamal Abdel Nasser University of Conakry, Conakry, Guinea, 3 Program Support, Management and Coordination Unit, Ministry of Health, Conakry, Guinea, 4 National HIV/AIDS and Hepatitis Control Program, Ministry of Health, Conakry, Guinea, 5 GeoHealth Group, Institute of Global Health, Faculty of Medicine, University of Geneva, Geneva, Switzerland, 6 Institute for Environmental Sciences, University of Geneva, Geneva, Switzerland, 7 Institute of Sea Sciences; Centro Superior de Investigaciones Científicas, Barcelona, Spain, 8 World Health Organisation, Geneva, Switzerland, 9 The Global Fund to Fight AIDS, Tuberculosis and Malaria, Geneva, Switzerland, 10 International Health Program; Catalan Institute of Health; University Hospital Germans Trias i Pujol, Badalona, Spain, 11 Foundation to Fight against AIDS and Infectious Diseases, Badalona, Spain, 12 Institut per la Recerca en Ciències de la Salut Germans Trias i Pujol, Badalona, Spain

* xvallesc.mn.ics@gencat.cat

**Data Availability Statement:** Data are accessed upon request and after approval of the competing ethical board. Data has been made available to

## Abstract

Few studies have investigated retention in HIV care in West Africa. We measured retention in antiretroviral therapy (ART) programmes among people living with HIV and re-engagement in care among those lost to follow up (LTFU) in Guinea and identified associated risk factors using survival analysis. Patient-level data were analysed from 73 ART sites. Treatment interruptions and LTFU were defined as missing a ART refill appointment by over 30 days and by over 90 days respectively. A total of 26,290 patients initiating ART between January 2018 and September 2020 were included in the analysis. The mean age at ART initiation was of 36.2 years, with women accounting for 67% of the cohort. Retention 12 months after ART initiation was 48.7% (95%CI 48.1–49.4%). The LTFU rate was 54.5 per 1000 person-months (95% CI 53.6–55.4), with the peak hazards of LTFU occurring after the first visit and decreasing steadily over time. In an adjusted analysis, the hazards of LTFU were higher among men compared to women (aHR = 1.10; 95%CI 1.08–1.12), being aged 13–25 years old versus older patients (aHR = 1.07; 95%CI = 1.03–1.13), and among those initating ART in smaller health facilities (aHR = 1.52; 95%CI 1.45–1.60). Among 14,683 patients with an LTFU event, 4,896 (33.3%) re-engaged in care, of whom 76% did so within six months from LTFU. The re-engagement rate was 27.1 per 1000 person-months (95%CI 26.3–27.9). Treatment interruptions were correlated with rainfall patterns and end of year

reviewers attached in a compressed file with data dictionary and basic instructions alongside an accessible URL. Data has been anonymized according to current rules.

**Funding:** This work was funded by the Global Fund to fight against AIDS, malaria and Tuberculosis (KK, CA, ST, FG, FA, PD, SA, FF, SS, DI, NN, MM, KA, KL and KY); Grant num. GIN-H-MOH. The funders had no role in study design, data collection and analysis, decision to publish, or preparation of the manuscript.

**Competing interests:** The authors have declared that no competing interests exist.

mobility patterns. Rates of retention and re-engagement in care are very low in Guinea, undermining the effectiveness and durability of first-line ART regimens. Tracing interventions and differentiated service delivery of ART, including multi-month dispensing may improve care engagement, especially in rural areas. Further research should investigate social and health systems barriers to retention in care.

## Introduction

Since 2002, antiretroviral therapy (ART) has been scaled up in low- and middle-income countries (LMIC). By 2019, an estimated 25.4 million (24.5–25.6 million) people living with HIV (PLHIV) were accessing ART globally [1], the majority of whom were residing in sub-Saharan Africa where the highest burden of HIV infection occurs. In 2013, UNAIDS launched the 90-90-90 targets, which aimed to have 90% of all PLHIV knowing their HIV status, 90% of diagnosed PLHIV receiving ART and 90% of those on ART for achieving viral suppression by 2020 [1]. However, several countries, particularly those in West Africa, have not met these targets, and renewed efforts will be needed if the 95-95-95 goals are to be met for 2030 [2].

Ensuring that patients receive and adhere to ART is critical for achieving optimal clinical outcomes, including reduced risks of HIV-related morbidity and mortality [3, 4]. Furthermore, sustained adherence to ART is necessary for patients to achieve viral load suppression which can reduce the risk of treatment failure, emerging drug resistance and onward HIV transmissions [5–9]. A cohort study of ART programmes in sub-Saharan Africa estimated retention in care among PLHIV on ART to be 67% at 5 years of follow-up, falling to around 50% at 5 years for West Africa [10]. A meta-analysis of studies from LMIC found that LTFU was highest among men, older patients, single persons, those who were unemployed, those with lower educational attainment, those with advanced WHO stage at initiation, those not having disclosed their HIV status, those not receiving cotrimoxazole prophylactic therapy when indicated, those receiving ART at a secondary level facility, and those with more recent year of ART initiation [11]. However, the underlying root causes of poor retention in care are likely to include a complex intersection between health systems, social and individual-level barriers, such as supply chain issues, stigma and lack of adequate follow-up or social support [12]. Additional environmental factors, such as poor accesability of roads during the rainy season, may also explain interrupted access to ART clinics at certain times during the year in some settings.

Loss to follow-up (LTFU) is a term that is used to cover an amalgamation of outcomes including death, default or not-documented transfer between ART clinics, and is usually defined as missing a scheduled ART clinic appointment by over 90 days with no documented cause [13]. In most analyses of ART retention, treatment interruptions of up to 90 days are generally not considered, leading to an under-estimate of the risk of treatment failure, and an over-estimate of the proportion of PLHIV who are effectively on ART. The majority of studies that have investigated risk factors for LTFU from ART services in sub-Saharan Africa have been undertaken in the Southern and Eastern region, with relatively little research in West and Central Africa [10], and no published data from Guinea. Most analyses have considered individual and clinic-level risk factors for LTFU, and few have assessed the role of environmental conditions such as rain patterns in farming subsistence economies. These seasonal factors may drive migration patterns or lead to individuals prioritizing economic activities over attending health facilities to obtain ART refills, particularly when distances are long to reach ART sites.

Furthermore, studies investigating re-engagement in care following an LTFU event are scarce in sub-Saharan Africa, despite the higher HIV transmission risks among people who interrupt ART [14]. These studies have shown a substantial number of patients experience treatment interruptions, with risk factors for re-engagement in care being similar to those associated with retention in care [15, 16].

The aim of this study was to estimate retention in antiretroviral therapy (ART) programmes among people living with HIV and re-engagement in care among those lost to follow up (LTFU) in Guinea and identified associated risk factors for these two outcomes. In addition, we assessed correlations between ART interruptions and rainfall seasonal patterns.

## Materials and methods

### Study setting

Guinea has a population of 11.6 million habitants living across a land mass of 245,857 km2 which varies from semi-arid climate in the north-east (i.e. the region of Kankan) to tropical in the south (i.e. the region of N'Zérékoré). Around 1.7 milion people live in the capital, Conakry. In 2019, HIV prevalence was estimated at 1.4% among 15–49 year olds [17], with higher prevalence observed among key populations, including men who have sex with men (11.4%) and female sex workers (10.7%) [17]. Of the estimated 110,000 (95%CI = 95,000–130,000) PLHIV in 2019, an estimated 57% (95% CI:49–66%) were on ART [17].

HIV treatment was first made available through the public sector in 2005, and was gradually decentralised thereafter, resulting in ART being available in 85 health facilities in eight administrative regions (Conakry, Boké, Farahnah, Mamou, Kindia, N'Zérékoré, Labé and Kankan) by the end of 2019, with around half of the PLHIV in care followed up in sites located in the capital Conakry. The "test and treat" strategy was adopted as national policy in 2017.

### Study design

A retrospective longitudinal analysis of the ART patient cohort was carried out using routine data obtained through the *Modèle Simplifié Reproductible* (MSR) software, a Microsoft Excel-based tool that was developed to monitor the ART patient cohort and ART stocks and implemented in health centres. For the purposes of this analysis, data were used from 73/85 (86%) sites (Fig 1), representing an estimated of 95% of the national ART patient cohort.

The dataset included visit level data, including sex, age at ART initiation, date of ART initiation, therapeutic regime and attendance dates for ART refills. ART refills were scheduled on a monthly basis. Visits are ascertained at site level and each patient has an unique identifier upon enrolement, but a new identifier was generated if a patient moves to another site without documentation or self-identification as PLVIH under ART. The follow-up period was from January 2018 until December 2020.

### Inclusion criteria

Patients were included in the analysis if they had ever initiated ART between January 2018 and September 2020 (therefore, excluding those with an observation period of less than 90 days) and if they were aged 13 years and over at the time of ART initiation. Patients initiating treatment before 2018 were excluded from the analysis, since the MSR was not fully implemented and the date of treatment initiation was not available for most of them.

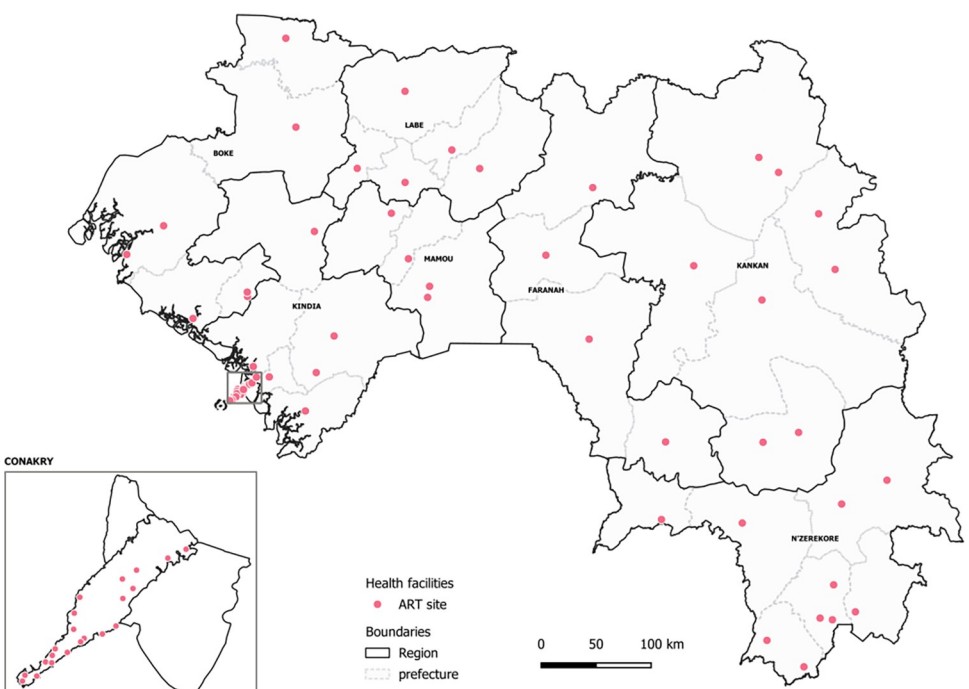

**Fig 1. Distribution of ART sites in Guinea included in the analysis (N = 73).** The base layer of the map could be obtained at https://gadm.org/download_country.html.

## Outcome definitions

A LTFU event was defined as not attending the scheduled ART refill for at least 90 consecutive days (three consecutively monthly refill appointments missed). Re-engagement was defined as a return to the cohort following a LTFU event. A treatment interruption was defined as not attending a scheduled ARV refill for more than 30 days. We also calculated the percentage of patients that did not attended their scheduled ART refill appointment for each month. Multi-month refills were taken into account when considering the LTFU start date.

## Data management and statistical methods

Data from the MSR were obtained in *csv* format from the 73 included ART sites and consolidated into *dta* and *R* format in a single dataset. Data cleaning was carried out and duplicates were removed. The cleaned data were analysed using the statistical packages Stata 14.0 software and R vs. 4.1.2.

Continuous variables were described using means and Standard Deviation (SD) after testing for normal distribution (Skewness and Kurtois test), and using medians and interquartile ranges (IQR) for skewed distributions. Proportions were described for categorical variables.

Kaplan-Meier curves were used to estimate i) the cumulative probability of patients who were retained in care at 6, 12 and 24 months following ART initiation and ii) the cumulative probability of re-engagement in care within 6 and 12 months after an LTFU event, stratified by sex and region. Smoothed hazard rates were estimated to identify peaks of LTFU and re-engagement. Rates of LTFU and re-engagement were assessed per 1000-person months of follow-up. Cox regression was used to identify crude and adjusted factors associated with LTFU and re-engagement using Hazard Ratios (HR) with corresponding 95% Confidence Intervals (CI), including clustering effect adjustments at the facility level. Cox models were adjusted for

clustering of patients in facilities using cluster-based robust standard errors within facilities. A p-value ≤0.05 was considered statistically significant.

Rainfall data were acquired from ERA5-Land reanalysis dataset [18] through the Copernicus Climate Change Service. ERA5-Land is a global land-surface atmospheric dataset with a spatial resolution of 9km spanning between 1981 to present. For the correlation analysis between rainfall and treatment interuptions, monthly means of precipitation data were used, averaged for each administrative region of Guinea. Both time series (precipitation and treatment interruption) were smoothed with spline interpolation method with a degree factor of 5. Linear correlation between the two variables was measured with Pearson's correlation coefficient.

### Ethical issues

Data were anonymized before the procurement of the MSR data sets. The study received ethical approval from the Comité National d'Éthique pour la Recherche en Santé (Ministry of Health from Guinea), with the reference number 139/CNERS/21. Consent to participants was waived by the Ethical Board due to the anonymity of the data sets used during the analysis.

## Results

### Participant characteristics

A total of 26,290 patients with unique identifiers were included in the retrospective analysis (in Table 1).

The median number of patients on ART across the 73 facilities was 873, with a range from 5 to 2,562. Overall, 17,690 (67.3%) participants were female and the overall mean age at ART initiation was 36.2 years (SD = 12.1), being lower among women at 34.0 years (SD = 11,5) compared to men [40.8 years (SD = 12.1; p<0.001)]. 12,293 (46.8%) of participants were followed-up in centers located in the capital Conakry. 10,548 (40.1.%) participants had initiated ART in 2018. 74.1% of patients were prescribed Lamiduvine+Tenofovir+Efavirenz (TDF+3TC+EFV).

### Cumulative probability of retention on ART

55.9% (N = 14,687) of patients experienced at least one LTFU event during the study period, with this being higher in men compared to women (59.7% vs. 54.0.%, p<0.001). The median time to LTFU was of 12 months (IQR 11–12). The cumulative probability of retention in care at 6 months after ART initiation was 65.9% (95%CI 65.3–66.5%), declining to 48.7% at 12 months (95%CI 48.1–49.4%) and 34.1% at 24 months (95%CI 33.4–34.8). The cumulative probability of retention was higher among women compared to men (Fig 2A) and differed by region (Fig 2B), with a higher probability of retention among PLHIV who initiated ART in a facility in Conakry region compared to those who initiated treatment elsewhere.

### Rates and risk factors for loss to follow up

The analysis considered 22,455 persons-years (PY) of follow-up during which 14,686 LTFU events occurred. The estimated rate of LTFU was 54.5 per 1000-person-months (95% CI 53.5–55.4), but varied by region, being lower in Conakry at 45.5 per 1000-person-months (95%CI 44.4–46.7; see Table 2).

Furthermore, rates of LTFU substantially differed across the regions outside of Conakry, ranging from 102.3 per 1000-person-months in Faranah (95%CI 94.5.-111.1) to 43.2 per 1000-person-months in Labe (95% CI 39.5–47.3) (Table 2). The peak hazards of LTFU was observed just after ART initiation, with 30.9% of all LTFU events occurring after the first visit,

**Table 1. Participant characteristics.**

| Category | Overall N | Overall % | 2018 N | 2018 % | 2019 N | 2019 % | 2020 N | 2020 % |
|---|---|---|---|---|---|---|---|---|
| *Total* | 26290 | 100 | 10548 | 40.1 | 8078 | 30.7 | 7664 | 29.2 |
| *Sex[1]* | | | | | | | | |
| Male | 8597 | 32.7 | 3461 | 32.8 | 2631 | 32.6 | 2505 | 32.7 |
| Female | 17690 | 67.3 | 7087 | 67.2 | 5444 | 67.4 | 5159 | 67.3 |
| *Age at initiation (years)* | | | | | | | | |
| 13–25 | 5322 | 20.2 | 2212 | 21.0 | 1578 | 19.5 | 1532 | 20.0 |
| 26–35 | 8911 | 33.9 | 3694 | 35.0 | 2611 | 32.3 | 2606 | 34.0 |
| 36–45 | 6538 | 24.9 | 2455 | 23.3 | 2181 | 27.0 | 1902 | 24.8 |
| >45 | 5299 | 20.2 | 2101 | 19.9 | 1609 | 19.9 | 1589 | 20.7 |
| Missing | 220 | 0.8 | 86 | 0.8 | 99 | 1.2 | 35 | 0.5 |
| Median (IQR) | 35 | (27–43) | 34 | (27–43) | 35 | (28–43) | 35 | (28–44) |
| *Facility level* | | | | | | | | |
| *Capacity quartiles [2]* | | | | | | | | |
| Lowest -25 (5–500; N = 44) | 6688 | 25,4 | 2334 | 22.1 | 1533 | 19.0 | 2413 | 31.5 |
| 25–50 (523–848; N = 11) | 6477 | 24,6 | 2596 | 24.6 | 2031 | 25.1 | 1838 | 24.0 |
| 50–75 (878–1294; N = 7) | 6672 | 25.4 | 2837 | 26.9 | 2224 | 27.5 | 1894 | 24.7 |
| 75–100 (1298–2613; N = 4) | 6453 | 24.6 | 2781 | 26.4 | 2290 | 28.4 | 1519 | 19.8 |
| NGO-supported | 6332 | 24.1 | 2601 | 24.7 | 1906 | 23.6 | 1825 | 23.8 |
| Government-run | 19958 | 75.9 | 7947 | 75.3 | 6154 | 76.4 | 5839 | 76.2 |
| *Region* | | | | | | | | |
| Conakry | 12293 | 46.8 | 5343 | 50.7 | 3542 | 43.9 | 3408 | 44.5 |
| Boké | 2770 | 10.5 | 965 | 9.2 | 973 | 12.1 | 832 | 10.9 |
| Kindia | 3338 | 12.7 | 1401 | 13.3 | 1070 | 13.3 | 867 | 11.3 |
| Labe | 975 | 3.7 | 303 | 2.9 | 396 | 4.9 | 276 | 3.6 |
| Mamou | 647 | 2.5 | 257 | 2.4 | 219 | 2.7 | 171 | 2.2 |
| Kankan | 2619 | 10.0 | 1059 | 10.0 | 769 | 9.5 | 791 | 10.3 |
| Farahnah | 883 | 3.7 | 204 | 1.9 | 334 | 4.1 | 345 | 4.5 |
| N'Zérékoré | 2765 | 10.5 | 1016 | 9.6 | 775 | 9.6 | 974 | 12.7 |

1. Three missing values. 2. The quartiles are defined such that the lowest quartile includes the facilities that encompass 25% of patients engaged in care during the study period. Into brackets we indicate the intervals of number of patients included in each quartile number and N represents the number of facilities included in each strata.

and steadily decreased thereafter (Fig 3A). In the crude analysis, LTFU was associated with male gender, being in care outside of Conakry region, being followed in facilities with the smaller patient cohort size and being enrolled in ART sites that were government-run (Table 2).

In the adjusted analysis, which included adjustements for clustering effects at facility level, the hazards of LTFU were higher among men compared to women (aHR = 1.10; 95% CI = 1.08–1.12), among those aged 13–25 years old compared to those aged 25 to 35 at initiation (aHR = 1.07; 95%CI = 1.02–1.13), among those initating ART in a health facility with a smaller patient cohort (aHR = 1.52; 95%CI = 1.0o-2.33).

## Cumulative probability of re-engagement in care

Among 14, 684 individuals who experienced a LTFU event included in this cohort, 9,554 (65.1%) were women and the mean age was 36.1 yrs. (SD = 12.2). 4,896 (33.3%) re-engaged in treatment during the study period. Among those who re-engaged, 76.4% did so during the first six months after the LTFU event (see Fig 3B). The cumulative probability of re-

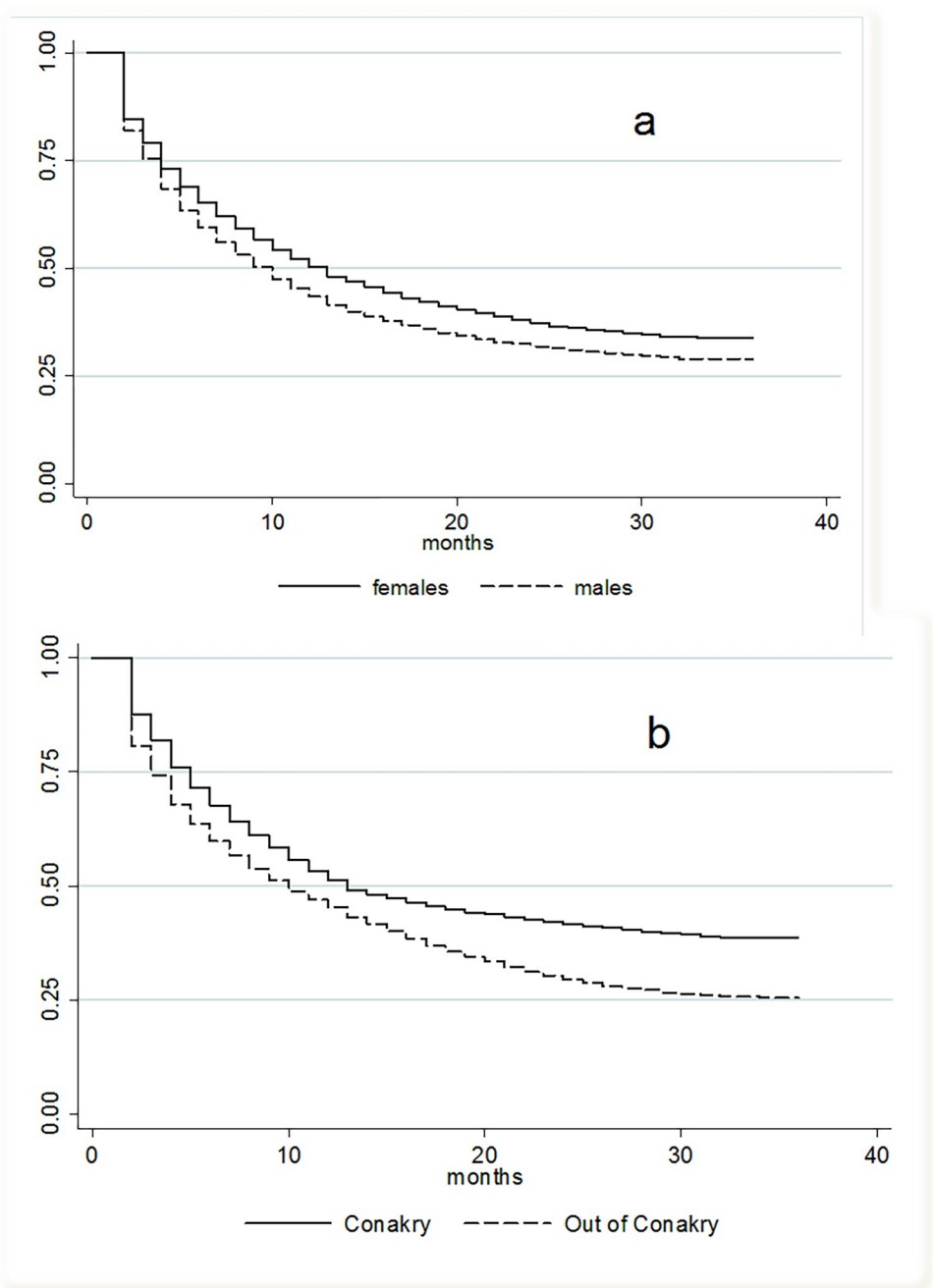

**Fig 2. a and b:** Kaplan-Meier estimates of cumulative probability of retention on ART by sex (a) and by region (b).

engagement was 27.1% at 6 months (95%CI = 26.3–27.9) and 35.2% at 12 months (95% CI = 34.4–36.1) after LTFU. Re-engagement rates were higher among females (Fig 4A) and among patients enrolled in ART sites outside of Conakry region (Fig 4B).

## Rates and risk factors for re-engagement in care

A total of 15,352 PY were included in the analysis and the re-engagement rate was 26.6 per 1000 person-months (95% CI: 25.8–27.3) (see Table 3).

**Table 2. Rates and crude and adjusted HR for LTFU, by baseline characteristics of patients.**

| Category | | | | | Crude | | | Adjusted [2] | | |
|---|---|---|---|---|---|---|---|---|---|---|
| *Sex* | PY | N of events | Rate[1] | 95%CI | HR | 95%CI | p | HR | 95%CI | p |
| Male | 6850 | 5129 | 62.4 | (62.4–64.1) | 1.09 | (1.07–1.11) | <0.001 | 1.10 | (1.08–1.12) | <0.001 |
| Female | 15604 | 9554 | 51.0 | (50.0–52.1) | 1 | — | — | 1 | — | — |
| *Year at initiation* | | | | | | | | | | |
| 2018 | 10651 | 8127 | 63.6 | (62.2–65.0) | 2.02 | (1.92–2.12) | <0.001 | 2.10 | (1.72–2.55) | <0.001 |
| 2019 | 7691 | 4306 | 46.7 | (45.3–48.1) | 1.34 | (1.28–1.42) | <0.001 | 1.42 | (1.17–1.72) | <0.001 |
| 2020 | 4113 | 2253 | 45.6 | (43.8–47.6) | 1 | — | — | 1 | — | — |
| *Region* | | | | | | | | | | |
| Out of Conakry[2] | 10884 | 8362 | 64.0 | (62.7–65.4) | 1.31 | (1.27–1.35) | <0.001 | 1.34 | (0.89–2.01) | 0.2 |
| Conakry | 1158 | 6324 | 45.5 | (44.4–46.7) | 1 | — | — | 1 | — | — |
| *Age* | | | | | | | | | | |
| 13–25 | 4473 | 3083 | 57.4 | (55.4–59.5) | 1.04 | (0.99–1.09) | 0.08 | 1.07 | (1.02–1.13) | 0.009 |
| 25–35 | 7592 | 4991 | 54.8 | (53.3–56.3) | 1 | — | — | 1 | — | — |
| 35–45 | 5744 | 3511 | 50.9 | (49.3–52.6) | 0.94 | (0.89–0.98) | 0.002 | 0.93 | (0.87–0.99) | 0.03 |
| >45 | 4440 | 2972 | 55.8 | (53.8–57.8) | 1.02 | (0.97–1.06) | 0.5 | 0.99 | (0.93–1.05) | 0.8 |
| *Size of facility (IQR)* | | | | | | | | | | |
| <25 | 4310 | 4376 | 84.6 | (82.1.-87.2) | 1.60 | (1.21–1.31) | <0.001 | 1.52 | (1.00–2.33) | 0.05 |
| 25–50 | 6278 | 2834 | 38.9 | (37.6–50.4) | 0.85 | (0.81–0.89) | <0.001 | 0.82 | (0.46–1.46) | 0.5 |
| 50–75 | 5376 | 3903 | 60.1 | (58.6–62.4) | 1.26 | (1.21–1.32) | <0.001 | 1.09 | (0.61–1.94) | 0.8 |
| >75 | 6490 | 3473 | 44.6 | (43.1–46.1) | 1 | — | — | 1 | — | — |
| *Facility type* | | | | | | | | | | |
| NGO-supported | 5535 | 3711 | 55.9 | (54.1–57.7) | 1.04 | (1.00–1.08) | 0.04 | 1.28 | (0.76–2.15) | 0.2 |
| Government-run | 16920 | 10975 | 54.1 | (53.1–55.1) | 1 | — | — | 1 | — | — |

1. Rates are expressed as 1000 person-months; 2. Rates for regions outside of Conakry were 45.5 (95%CI = 46.0–48.6) for Boke, 102.3 (95%CI = 94.5–111.1) for Faranah, 107.4 (95%CI = 102.8–112.3.) for Kankan, 62.4 (95%CI = 59.8–65.1 for Kindia), 43.2 (95%CI = 39.5–47.3) for Labe, 58.0 (95%CI = 52.3–64.4) for Mamou and 57.3 (95% CI = 54.5–60.2) for N'Zérékoré. 2. Adjusted by all study variables and clustering effects within facilities.

The crude analysis showed that being female, being followed up outside Conakry, being enrolled in facilities with larger patient cohorts, and being enrolled in government-run facilities were associated with higher rates of re-engagement. In the adjusted analysis, which included adjustments for clustering effects at the facility level, the same variables remained associated with higher rates of re-engagement: the hazards of re-engagement were higher among females compared to males (aHR = 1.16; 95%CI = 1.08–1.24), among those being followed in facilities outside of Conakry region (aHR = 2.02; 95%CI = 1.08–3.79), and those enrolled in facilities with larger patient cohorts (aHR = 3.17; 95%CI = 1.47–6.84), but not among those enrolled in government-run facilities.

## Retention in care seasonality and rainfall patterns

Trends in ART treatment interruptions and LTFU demonstrated a clear seasonal pattern, showing a negative correlation with rainfall (Pearson's coefficient of -0.57) in the regions outside of Conakry (Fig 5A) with a peak in treatment interruptions from May to September during 2018 and 2019. In contrast, we found a weak correlation between rainfall and treatment interruptions in the Conakry region (Pearson's coefficient of -0.06, Fig 5B).

A peak of treatment interruptions was consistently observed in January across Guinea in 2019 (23.2%) and 2020 (23.0%), alongside the peaks in September 2018 (18.5%) and 2019 (19.2%), at the end of the rainy season outside of Conakry (Fig 5A). However, 68.6%, 49.1%,

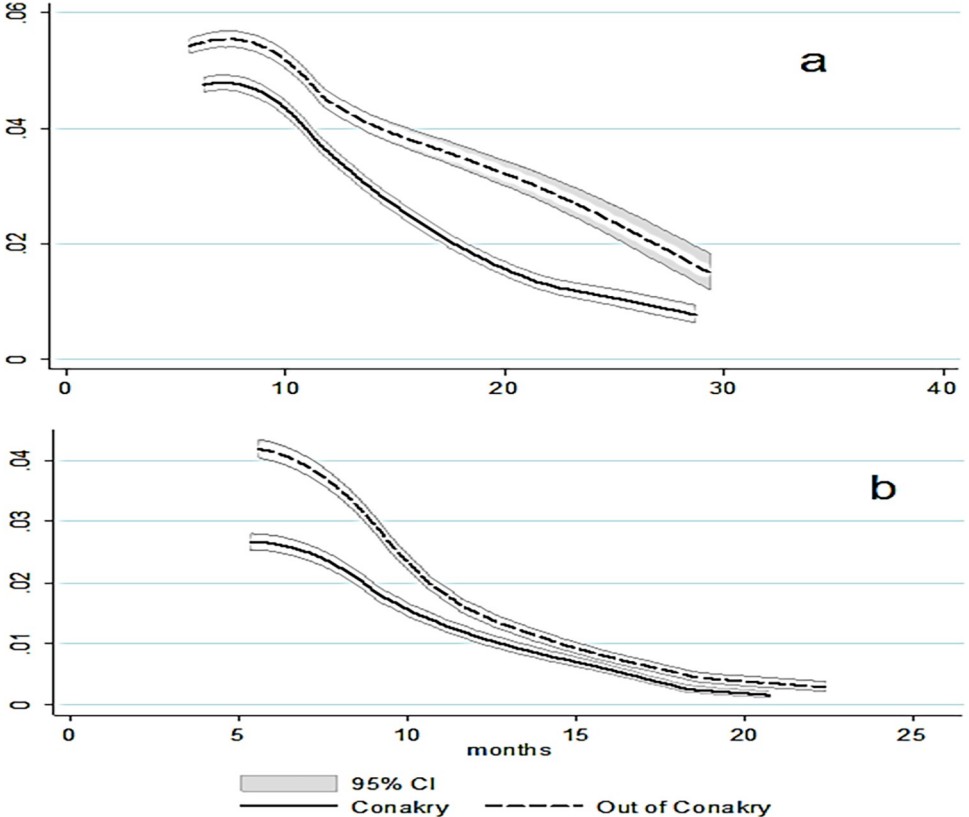

**Fig 3. a and b:** Smoothed hazard estimates of LTFU (a) and re-engagement (b) by year, stratified by region.

75.5% and 79.4% of those interrupting treatment returned to the cohort within three months, respectively.

## Discussion

Our longitudinal analysis showed worryingly poor rates of retention in care among ART patients who enrolled in the Guinean national HIV treatment programme between January

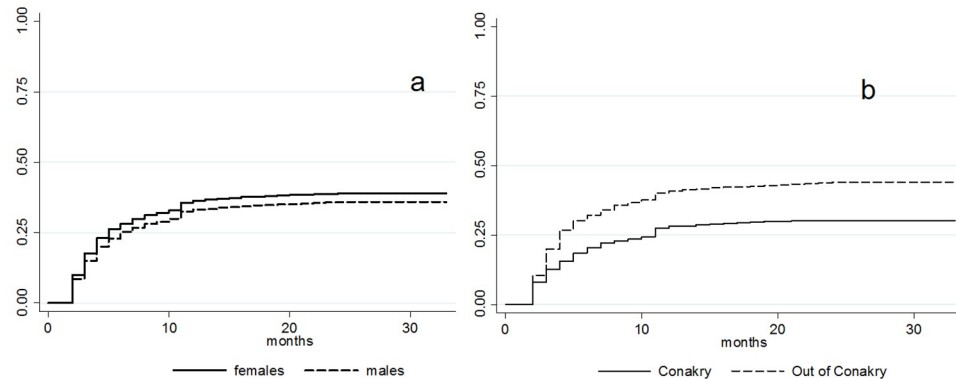

**Fig 4. a and b:** Kaplan-Meier estimates of the cumulative probability of re-engagement to ART by sex (a) and by region (b).

**Table 3. Rates and crude and adjusted HR for re-engagement by baseline characteristics of patients.**

| Category | | | | | Crude | | | Adjusted [2] | | |
|---|---|---|---|---|---|---|---|---|---|---|
| *Sex* | *PY* | *N of events* | *Rate* | *(95%CI)[1]* | HR | 95%CI | p | HR | 95%CI | p |
| Male | 5501 | 1608 | 24.4 | (23.2–25.6) | 1 | — | <0.001 | 1 | — | — |
| Female | 9846 | 3288 | 27.8 | (26.9–28.9) | 1.12 | (1.06–1.19) | | 1.17 | (1.09–1.25) | <0.001 |
| Year of initiation | | | | | | | | | | |
| 2018 | 10828 | 3001 | 23.1 | (22.3–23.9) | 1.10 | (0.99–1.22) | 0.09 | 1.04 | (0.81–1.23) | 0.5 |
| 2019 | 3721 | 1487 | 33.3 | (31.7–35.0) | 1.15 | (1.03–1.29) | 0.01 | 1.00 | (0.82–1.30) | 0.9 |
| 2020 | 780 | 408 | 43.6 | (39.6–48.1) | 1 | — | — | 1 | — | — |
| Out of Conakry[2] | 7889 | 3193 | 33.7 | (32.6–34.9) | 1.60 | ( 1.51–1.70) | <0.001 | 2.32 | (1.10–4.89-) | 0.03 |
| Conakry | 7463 | 1703 | 19.0 | (18.1–19.9) | 1 | — | — | — | — | — |
| *Age (years)* | | | | | | | | | | |
| 13–25 | 5204 | 756 | 25.9 | (24.1–27.8) | 0.97 | (0.89–1.05) | 0.4 | 0.96 | (0.88–1.04) | 0.3 |
| 25–35 | 5068 | 1631 | 26.8 | (25.5–28.2) | 1 | — | — | 1 | — | — |
| 35–45 | 3936 | 1345 | 28.5 | (27.0–30.0) | 1.05 | (0.98–1.13) | 0.2 | 1.04 | (0.92–1.17) | 0.5 |
| >45 | 3835 | 1162 | 25.2 | (23.8–26.7) | 0.95 | (0.88–1.03) | 0.2 | 0.96 | (0.88–1.03) | 0.3 |
| *Size of facility (IQR)* | | | | | | | | | | |
| <25 | 4472 | 870 | 16.2 | (15.2–17.3) | 1 | — | — | 1 | — | — |
| 25–50 | 3828 | 1089 | 23.7 | (22.3–25.2) | 1.43 | (1.31–1.56) | <0.001 | 1.56 | (1.07–2.26) | 0.02 |
| 50–75 | 3794 | 1183 | 26.0 | (24.5–27.5) | 1.58 | (1.45–1.72) | <0.001 | 1.82 | (1.02–3.25) | 0.04 |
| >75 | 3258 | 1754 | 44.9 | (42.8–47.0) | 2.51 | (2.32–2.73) | <0.001 | 3.17 | (1.47–6.84) | 0.003 |
| *Facility type* | | | | | | | | | | |
| NGO-supported | 4029 | 1002 | 20.7 | (19.5–22.1) | 1 | — | — | — | — | 0.8 |
| Government-run | 11323 | 3894 | 28.7 | (27.8–29.6) | 1.33 | (1.24–1.42) | <0.001 | 1.10 | (0.54–2.20) | |

1. Rates are expressed per 1000 person-months. 2. Rates for regions outside of Conakry were of 34.3 (95%CI = 31.6–37.4) for Boke, 17.4 (95%CI = 14.5–20.8) for Faranah, 29.8 (95%CI = 27.6–32.1) for Kankan, 56.6 (95%CI = 53.5–60.0) for Kindia, 27.1 (95%CI = 23.1–31.8) for Labe, 19.8 (95%CI = 16.2–24.2) for Mamou and 24.1 (95%CI = 22.0–26.5) for N'Zérékoré. 2. Adjusted by all study variables and clustering effects within facilities.

2018 and September 2020, with a 12-month retention rate of 42.1% among men and 51.0% among women. These rates are notably lower than the 76.8% observed elsewhere in Africa [10], and in the neighboring country of Mali where it reached 84.3% [19]. The peak hazard of LTFU occurred after the first visit, in line with findings from elsewhere in sub-Saharan Africa [20–24]. The very high rates of attrition from ART in this setting were tempered by reasonable rates of re-engagement in care, with 35.2% of patients who had experienced LTFU being back on ART within 12 months after a LTFU event. However, the rate of re-engagement was consistent with findings from Zambia where it was 51.4% (95% CI = 33.2–72.5%) [15] and similar to the observed rate in Mali (39.0%) [16]. The poor retention in care and regular treatment interruptions observed in our study are of great concern, given the associated risks of treatment failure [5–8, 25], development of ARV resistance [4–6, 25, 26], and increased transmission of HIV drug-resistant strains.

Our analysis showed that predictors of LTFU were similar to those for re-engagement in care, as observed in other recent studies in the region [10, 15, 16, 19], with the exception of being followed in an urban facility [15]. This suggest that drivers of retention differ in rural and urban areas. As demonstrated in several other African studies, we also found higher rates of LTFU among men and after the first visit [10, 27, 28]. We also found that young patients aged 13 to 25 years old experienced higher rates of LTFU and lower re-engagement rates compared to older patients in line with previous studies in sub-Saharan Africa [29], possibly explained by greater HIV-related stigma [30], and lesser ability or resources to access ART

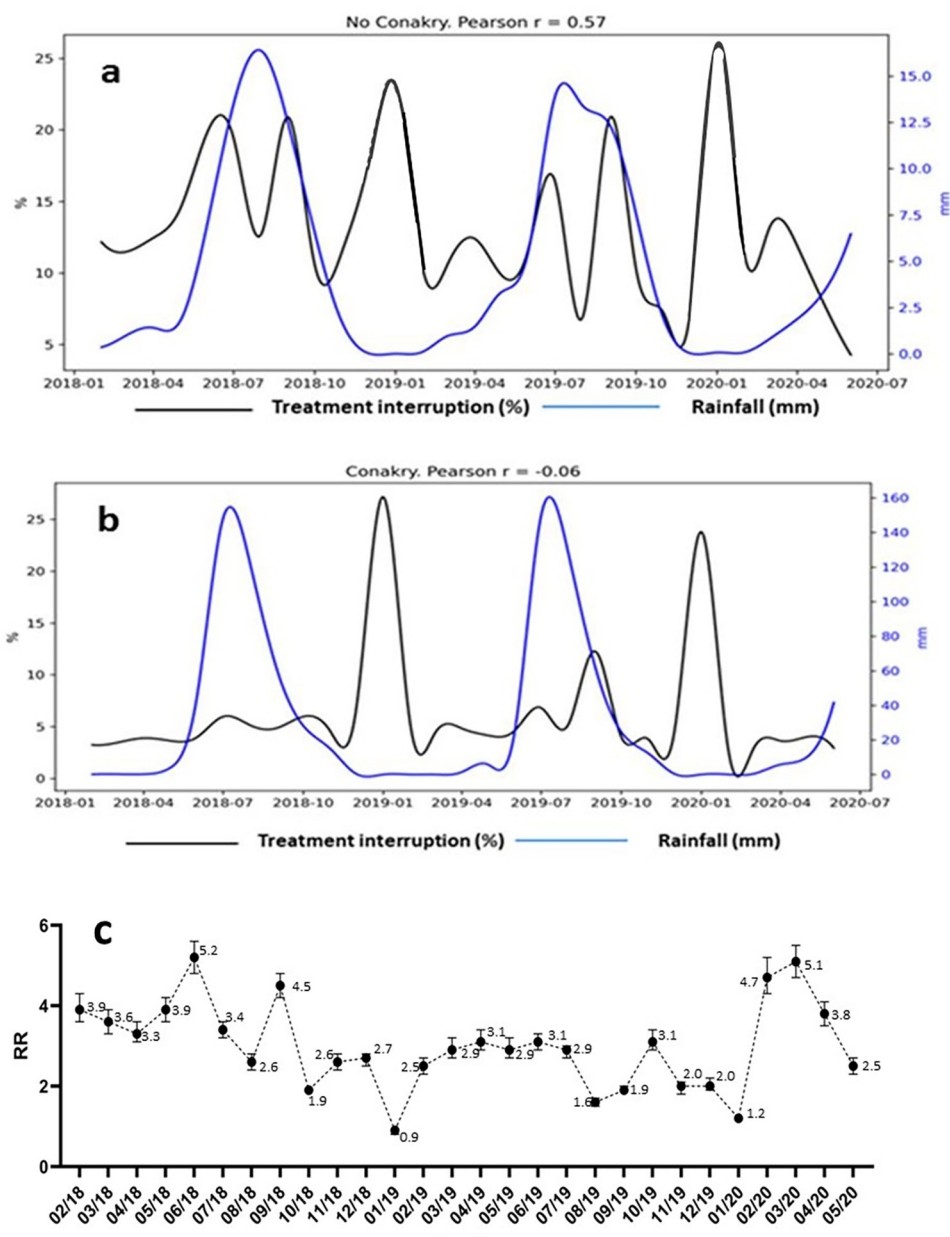

**Fig 5. a-c:** Rainfall pattern and proportion of treatment interruptions in regions out of Conakry and Conakry region (5a and 5b) RR of treatment interruptions by month of follow-up (5c)[1,2,3,4]. 1. The Y axis on the lefthand side shows the proportion of treatment interruptions, defined as the percentage of patients with scheduled appointments that did not attend the clinic, and the righthand the average rainfall in the entire country. 2. Data includes the period spanning between January 2018 to May 2020. 3. Includes all patients which had at last one registered visit during the study period. 4. RR of proportion of treatment interruptions between patients followed-up out of Conakry and in Conakry region.

clinics. Interventions tailored to young people living with HIV have not been implemented in Guinea, but have shown great promise in other countries and could be adapted to the Guinean context [31]. Finally, important regional differences in retention to care were observed. These may be explained by various factors, including those relating to cultural and religious beliefs, climatic differences (from semi-arid in Kankan to tropical in N'Zérékoré) and consequent

patterns of farming, the presence of informal mining in certain regions (i.e. Kankan which shows the highest rate of LTFU) which attract highly mobile populations, the existence of PLHIV associations, stock-outs and distance to quality ART care provision.

We found some evidence of clustering at the facility level in terms of region, size and facility type, suggesting some differences in the effectiveness of service delivery in relation to these factors, which denotes most probably an important influence of the quality of the service provided in each setting. In contrast, the clustering effect was barely observed for sex and age (similar CI between when including or excluding clustering adjustements), which denotes that these individual-level factors are independent of the quality of service provided (i.e. there is no discrimination over gender). Further qualitative research is merited to elicit social and health systems factors that influence retention in care in this setting, including the quality of services and cultural factors.

The poorer retention and lower re-engagement among patients attending NGO-supported clinics may be due to a higher proportion of advanced stage patients being referred there, resulting in higher mortality rates. The observed associations between rainfall patterns and treatment interruptions outside of the region of Conakry suggest that conditions related to climatic factors undermine access to treatment sites and should be further evaluated. Higher rates of treatment interruptions aligned with periods of increased farming activities that occur just before the peak of the rainy season. During the *soudure* period, before the new harvest, resources stored after the last farming season are almost gone, and it may be that PLHIV are forced to make a difficult choice between reaching ART centres and continuing to undertake farming activities. Furthermore, Guinea's road infrastructure is poor, with conditions exacerbated during the rainy season which may play a synergistic role in decreasing access [32]. Food insecurity and lack of economic resources for transportation to ART sites has been associated with LTFU in other sub-Saharan African countries [33–35]. These observations underscore the sensitivity of retention in care to economic crises, as well as droughts and floods associated with climate change [36]. The increase in the rate of treatment interruptions at the end of the year may be explained by cultural norms that encourage the population to return to their birth home during the end-of-year period for celebrations (PNLSH, personal communication). Analyses of the association between rainfall in the catchment area of a clinic and the proportion of patients who missed a clinic visit at this particular clinic were more complicated to interpret because other unassessed factors could explain these differences including local-level ARV stock-outs, cultural practices, migratory patterns and service quality, some of which are also linked to climatic patterns.

The underlying factors should be addressed through the introduction of multi-month dispensing of ART where stock levels permit. In addition, a comprehensive package of measures to improve ART retention could include nutritional support to food insecure patients, tracing activities, pshychosocial support, especially following ART initiation and during months where attrition tends to peak.

There are various limitations that need to be taken into account when interpreting our findings. Firstly, as in other countries with weak tracing and vital registration systems, the proportion of patients LTFU who had in fact self-transferred to another clinic or died could not be determined in our study. Our LTFU rate is likely over-estimated as studies from other contexts at a similar stage of scaling up ART have estimated that over 20% of patients LTFU patients had in fact died [10]. Furthermore, undocumented transfers to another clinic lead to an over-estimate of the number of PLHIV who have ever initiated ART, and under-estimates of retention in care rates but would lead to over-estimates of the number of PLHIV on ART as well as under-estimates of retention in care rates [37–40]. Similarly, we were unable to estimate mortality outcomes and so we could not distinguish between patients who had defaulted but

remained alive, and those who had died. Efforts to strengthen the tracing system, including follow up calls and visits once scheduled appointments are missed, would improve retention and also enable more accurate estimates of the effectiveness of the ART programme to be generated. Furthermore, implementing procedures to facilitate inter-clinic transfers would also improve the accuracy of estimates of the number of PLHIV who are LTFU.

In conclusion, our study demonstrated high levels of LTFU and re-engagement in care, undermining the effectiveness and durability of first-line ART regimens. Tracing systems and clinic transfer procedures should be strengthened to more accurately ascertain the number of deaths and self-transfers which would enable the true number of persons LTFU to be determined. Nevertheless, even if LTFU rates are over-estimated, the rates of retention in care are sub-optimal in this setting as demonstrated by the high rate of re-engagement in care following treatment interruptions. Several studies have shown that multi-month dispensing and other differentiated service delivery models of ART [41] may improve care engagement, especially in rural areas [42, 43]. These interventions are slowly being introduced in Guinea, and subsequent evaluations of retention in care will be useful for further refining their application in this context. A promising strategy would be the use of long-acting injectable ART which may reduce HIV treatment barriers with monthly or bi-monthly administration. However, evidence about feasibility and implementation considerations in LMICs is still lacking [44].

## Supporting information

**S1 Fig. Relative Risk of treatment interruptions between patients followed-up out in facility sites out of Conakry vs. Conakry region by month of follow-up (from February 2018 to May 2020)[1].** [1]We included facilities with the larger cohorts (>300 patients) from each patients. Faranah region was excluded due the low numbers included in each site.
(TIFF)

**S1 Data.**
(ZIP)

## Acknowledgments

We grateful acknowledge the ART health providers for their valuable support to PLHIV, and the PLNSH team.

## Author Contributions

**Conceptualization:** Kadio Jean-Jacques Olivier Kadio, Cissé Amadou, Saidou Diallo Thierno, Foromo Guilavogui, Fapeingou Tounkara Adrien, Pe Damey, Sow Alhassane, Fily Bah Fatoumata, Sékou Youla Souleymane, Diallo Ibrahima, Nestor Leno Niouma, Mboungou Lazare, Nyawotope Koffi Ahiatsi Arnold, Kaba Laye, Sy Zeynabou, Wringe Alison, Hoibak Sarah, Koïta Youssouf, Xavier Vallès.

**Data curation:** Kadio Jean-Jacques Olivier Kadio, Saidou Diallo Thierno, Foromo Guilavogui, Fapeingou Tounkara Adrien, Pe Damey, Sow Alhassane, Fily Bah Fatoumata, Sékou Youla Souleymane, Diallo Ibrahima, Nestor Leno Niouma, Mboungou Lazare, Kaba Laye, Sy Zeynabou, Wringe Alison, Hoibak Sarah, Xavier Vallès.

**Formal analysis:** Sy Zeynabou, Vallès-Casanova Ignasi, Wringe Alison, Xavier Vallès.

**Funding acquisition:** Nyawotope Koffi Ahiatsi Arnold, Hoibak Sarah, Koïta Youssouf.

**Investigation:** Kadio Jean-Jacques Olivier Kadio, Cissé Amadou, Saidou Diallo Thierno, Foromo Guilavogui, Sow Alhassane, Nyawotope Koffi Ahiatsi Arnold, Sy Zeynabou, Vallès-Casanova Ignasi, Hoibak Sarah, Koïta Youssouf, Xavier Vallès.

**Methodology:** Kadio Jean-Jacques Olivier Kadio, Cissé Amadou, Saidou Diallo Thierno, Sy Zeynabou, Xavier Vallès.

**Project administration:** Kadio Jean-Jacques Olivier Kadio, Cissé Amadou, Hoibak Sarah, Koïta Youssouf.

**Resources:** Koïta Youssouf.

**Supervision:** Kadio Jean-Jacques Olivier Kadio, Pe Damey, Wringe Alison, Hoibak Sarah, Koïta Youssouf.

**Validation:** Kadio Jean-Jacques Olivier Kadio, Kaba Laye, Wringe Alison, Hoibak Sarah, Koïta Youssouf, Xavier Vallès.

**Writing – original draft:** Kadio Jean-Jacques Olivier Kadio, Cissé Amadou, Saidou Diallo Thierno, Foromo Guilavogui, Fapeingou Tounkara Adrien, Pe Damey, Sow Alhassane, Fily Bah Fatoumata, Sékou Youla Souleymane, Diallo Ibrahima, Nestor Leno Niouma, Mboungou Lazare, Nyawotope Koffi Ahiatsi Arnold, Kaba Laye, Sy Zeynabou, Vallès-Casanova Ignasi, Wringe Alison, Hoibak Sarah, Koïta Youssouf, Xavier Vallès.

**Writing – review & editing:** Cissé Amadou, Saidou Diallo Thierno, Foromo Guilavogui, Fapeingou Tounkara Adrien, Pe Damey, Sow Alhassane, Fily Bah Fatoumata, Sékou Youla Souleymane, Diallo Ibrahima, Nestor Leno Niouma, Mboungou Lazare, Nyawotope Koffi Ahiatsi Arnold, Kaba Laye, Sy Zeynabou, Vallès-Casanova Ignasi, Wringe Alison, Hoibak Sarah, Koïta Youssouf.

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
