## [Decision Letter · Decision Letter 0]

10 Oct 2022

PGPH-D-22-01123

Retention in care among people living with HIV in the national antiretroviral therapy programme in Guinea: a retrospective cohort analysis

Dear Dr. Vallès,

Thank you for submitting your manuscript to PLOS Global Public Health. After careful consideration, we feel that it has merit but does not fully meet PLOS Global Public Health’s publication criteria as it currently stands. Therefore, we invite you to submit a revised version of the manuscript that addresses the points raised during the review process.

Please note that we have only been able to secure a single reviewer to assess your manuscript. We are issuing a decision on your manuscript at this point to prevent further delays in the evaluation of your manuscript. Please be aware that the editor who handles your revised manuscript might find it necessary to invite additional reviewers to assess this work once the revised manuscript is submitted. However, we will aim to proceed on the basis of this single review if possible. 

Could you please carefully revise the manuscript to address all comments raised?

We look forward to receiving your revised manuscript.

Kind regards,

Steve Zimmerman, PhD

PLOS Staff Editor

Journal Requirements:

1. Please send a completed 'Competing Interests' statement, including any COIs declared by your co-authors. If you have no competing interests to declare, please state "The authors have declared that no competing interests exist". Otherwise please declare all competing interests beginning with the statement "I have read the journal's policy and the authors of this manuscript have the following competing interests:"

3. Please provide separate figure files in .tif or .eps format only and remove any figures embedded in your manuscript file. Please also ensure that all files are under our size limit of 10MB.

4. We do not publish any copyright or trademark symbols that usually accompany proprietary names, eg ©, ®, or ™  (e.g. next to drug or reagent names). Please remove all instances of trademark/copyright symbols throughout the text, including ® on page 9.

5. We have noticed that you have uploaded Supporting Information files, but you have not included a list of legends. Please add a full list of legends for your Supporting Information files after the references list. 

6. In the online submission form, you indicated that "Data are accessed upon request and after approval of the competing ethical board. Data has been made available to reviewers attached in a compressed file with data dictionary and basic instructions alongised an accessible URL. Data has been anonymized according to current rules". All PLOS journals now require all data underlying the findings described in their manuscript to be freely available to other researchers, either 1. In a public repository, 2. Within the manuscript itself, or 3. Uploaded as supplementary information.

Additional Editor Comments (if provided):

Reviewers' comments:

Reviewer's Responses to Questions

**Comments to the Author**

1. Does this manuscript meet PLOS Global Public Health’s publication criteria? Is the manuscript technically sound, and do the data support the conclusions? The manuscript must describe methodologically and ethically rigorous research with conclusions that are appropriately drawn based on the data presented.

Reviewer #1: Partly

2. Has the statistical analysis been performed appropriately and rigorously?

Reviewer #1: Yes

3. Have the authors made all data underlying the findings in their manuscript fully available (please refer to the Data Availability Statement at the start of the manuscript PDF file)?

Reviewer #1: Yes

4. Is the manuscript presented in an intelligible fashion and written in standard English?

Reviewer #1: Yes

5. Review Comments to the Author

Reviewer #1: Peer-review report of the manuscript “Retention in care among people living with HIV in the national antiretroviral therapy programme in Guinea: a retrospective cohort analysis” (PGPH-D-22-01123).

Kadio and colleagues did a retrospective cohort study of rates and predictors of loss to follow-up and re-engagement in Guinea’s national ART program. The study reports concerningly low retention rates. The study also assessed associations between rainfall patterns and missed clinic visits and showed interesting correlations.

The manuscript is overall well written. Statistical analyses of rates and predictors of loss to follow-up and re-engagement are sound. However, analyses of associations between rainfall patterns and missed visits could be improved.

The most important limitation of the study is that the authors did not adjust loss to follow-up rates for undocumented mortality and silent transfer. Therefore, rates of loss to follow-up are likely to be substantially overestimated. I suggest the authors consider the likely overestimation of loss to follow-up rates in their conclusions.

I have the following specific suggestions for improvement:

1. Analyses of rates and predictors of loss to follow-up and re-engagement:

a. Please clarify in the Methods section how you adjusted Cox models for the clustering of patients in facilities. Did you use cluster-based robust standard errors?

b. Adjusted model A could be removed from Tables 2 and 3. Analysis should be adjusted for the clustering of patients in facilities. Model A (adjusted for patient characteristics but not for clustering) is unnecessary.

c. Please confirm the correctness of the results in Tables 2 and 3. If robust standard errors were used to adjust for clustering, I would expect that additional adjustment for clustering does not affect the point estimates but widens the confidence intervals. This is also the case for some but not for all estimates. For example, in Table 2, the adjusted hazard ratio for ART initiation for the year 2008 in model A is 2.10 (95 CI 2.00-2.20). In model B (additionally adjusted for clustering), the HR is the same as in model A, and the confidence interval is wider (aHR 2.10 95% CI 1.72-2.55). However, for patient characteristics (e.g. age and sex in Table 2), the point estimates and confidence intervals are identical in models A and B. Surprisingly, Table 3 point estimates for age, sex and out of Conakry are not identical in models A and B. Please double-check these results.

d. Conclusions regarding loss to follow-up rates should take into account likely overestimation due to lack of adjustment for silent transfer and undocumented mortality.

e. The data presented in the manuscript does not support the conclusion that tracing interventions, multi-month dispensing and other age and gender-specific interventions may improve care engagement. These interventions can be mentioned in the Discussion but should not be presented as Conclusions of this study.

2. Analyses of associations between rainfall patterns and ART interruptions:

a. The analysis of rainfall patterns and ART interruptions is novel and topical and could be presented more prominently.

b. The authors evaluated correlations between the average proportion of patients who missed a clinic visit outside of Conakry (5a) and in Conakry (5b) and the average rainfall in the entire country. Could this association be assessed on a facility level? I.e. the association between rainfall in the catchment area of a clinic and the proportion of patients who missed a clinic visit at this particular clinic.

c. Why are the peaks in missed visits in January 2019 and 2020 in panel 5a marked with a dashed line? Please clarify in the figure caption.

d. The authors calculated Pearson’s correlation coefficient (r) to assess associations between rainfall patterns and missed visits but did not evaluate the statistical uncertainty of this association. The authors could use more advanced statistical methods to calculate odds or risk ratios with 95% confidence intervals for associations between rainfall patterns and missed visits on the facility level. For example, case-crossover designs are widely used to estimate associations between time-varying environmental exposures and health outcomes from daily time series data (see https://bmcmedresmethodol.biomedcentral.com/articles/10.1186/1471-2288-14-122). Such an analysis might be beyond the scope of this study and could be presented in a separate manuscript.

Minor:

3. In the Abstract, the authors state that “patients initiating treatment between January 2018 and December 2020 were included” but in the Methods “ patients were included in the analysis if they had ever initiated ART between January 2018 and September 2020”. Please correct the end of the eligibility period in the Abstract or the Methods section.

4. Reference 10 is not a review but a cohort study

5. Please correct the number of included sites: in the Methods section, you state data from 73 facilities were used, but in the Abstract and Figure 1, you state 66 facilities were included.

6. Line 168: Were scheduled appointment dates recorded in the database? Please clarify.

7. The following sentence is unclear and should be revised: “Patients under differential treatment with frequency refills, each refill was accounted for 2 or more consecutive months of ARV refill as appropriate.”

8. I cannot follow the argument presented in lines 389-395 “The notorious effect of adjusting for clustering at facility level…”). Please rewrite this section.

6. PLOS authors have the option to publish the peer review history of their article (what does this mean?). If published, this will include your full peer review and any attached files.

**Do you want your identity to be public for this peer review?** For information about this choice, including consent withdrawal, please see our Privacy Policy.

Reviewer #1: No

---

## [Decision Letter · Decision Letter 1]

28 Feb 2023

PGPH-D-22-01123R1

Retention in care among people living with HIV in the national antiretroviral therapy programme in Guinea: a retrospective cohort analysis

Dear Dr. Vallès,

Thank you for submitting your manuscript to PLOS Global Public Health. After careful consideration, we feel that it has merit but does not fully meet PLOS Global Public Health’s publication criteria as it currently stands. Therefore, we invite you to submit a revised version of the manuscript that addresses the points raised during the review process.

We look forward to receiving your revised manuscript.

Kind regards,

Miquel Vall-llosera Camps

Staff Editor

Journal Requirements:

2. Please insert an Ethics Statement at the beginning of your Methods section, under a subheading 'Ethics Statement'. It must include:

a) (for human participants/donors) - A statement that formal consent was obtained (must state whether verbal/written) OR the reason consent was not obtained (e.g. anonymity). NOTE: If child participants, the statement must declare that formal consent was obtained from the parent/guardian.

Additional Editor Comments:

Thank you for submitting the revised version of your manuscript. The previous reviewer was not available to comment again on your revision. We invited two additional reviewers to assess your revised manuscript. The reviewers raised minor comments that need to be addressed.

Reviewers' comments:

Reviewer's Responses to Questions

**Comments to the Author**

1. If the authors have adequately addressed your comments raised in a previous round of review and you feel that this manuscript is now acceptable for publication, you may indicate that here to bypass the “Comments to the Author” section, enter your conflict of interest statement in the “Confidential to Editor” section, and submit your "Accept" recommendation.

Reviewer #2: (No Response)

Reviewer #3: (No Response)

2. Does this manuscript meet PLOS Global Public Health’s publication criteria? Is the manuscript technically sound, and do the data support the conclusions? The manuscript must describe methodologically and ethically rigorous research with conclusions that are appropriately drawn based on the data presented.

Reviewer #2: Yes

Reviewer #3: Yes

3. Has the statistical analysis been performed appropriately and rigorously?

Reviewer #2: Yes

Reviewer #3: Yes

4. Have the authors made all data underlying the findings in their manuscript fully available (please refer to the Data Availability Statement at the start of the manuscript PDF file)?

Reviewer #2: Yes

Reviewer #3: Yes

5. Is the manuscript presented in an intelligible fashion and written in standard English?

Reviewer #2: Yes

Reviewer #3: Yes

6. Review Comments to the Author

Reviewer #2: The primary purpose of the paper was to fully document poor rates of retention in care among ART patients in Guinea’s national HIV treatment program with exploration as to possible causes of poor treatment retention through analysis of patient demographics, clinic specific factors, and a unique look at rainfall and other season agricultural factors that might influence ART continuation. The authors did an admirable effort to document the poor ART retention and included in the limitations section of the manuscript the issue of lack of records of patient mortality as well as lack of records regarding patient’s receiving ART care in other facilities other than the numerous clinics included in this study. Documentation of ART adherence and clinic retention is an important area of evaluation and research, particularly for those nations that fall below continental averages for ART adherence, which was the situation for Guinea.

The literature review was focused on the issues the current paper wished to address and established that retention in HIV care/ART adherence is a challenge for multiple medical care systems. The authors pointed out that the retention rates in Guinea at the clinics in question was lower than elsewhere in Africa, including in neighboring Mali. The literature review in the introduction set the stage for the problem faced by medical providers in Guinea.

Data collection and analysis was well documented and supported the claim of low ART adherence rates among multiple clinical providers. There was analysis that indicated the low ART adherence rate was higher among rural smaller clinics than larger urban clinics. But overall, the point was well established in their analysis that low ART adherence is a problem. There appeared to be no deviations from the proposed data collection and analysis plan. The authors point out that the rainfall data did not have the specificity for definitive analysis but was adequate for the conclusions they derived regarding ART adherence.

The paper is suitable for publication but there are a few typographical errors that need

to be addressed. These include changing the word ‘African’ to ‘Africa’ in line 52; the correct word is ‘amalgamation’ in line 111; a comma was used instead of a decimal mark so that ‘1,7’ should be changed to ‘1.7’ on line 145; ‘rug-resistant’ should be changed to ‘drug-resistant on line 360; on line 373 consider whether ‘regionals’ or ‘regional’ is the best word to use; and online 420 the word ‘psychosocial’ is the correct term.

The primary concern of this reviewer is with the final recommendations of multi-month ART dispersing and targeted ART adherence strategies for men and for younger HIV positive persons. The authors clearly state the problems with treatment failure and treatment resistance that may take place with sporadic ART adherence. However, their recommendations are not complete based upon the current science of ART adherence. Injectable ART for 30 days should have been considered in the conclusion section. Whereas multi-month ART dispersing would certainly be a sound recommendation, emerging research on 30-day and potentially 60-day ART injections may offer those HIV patients ART treatment who cannot travel during rainy season or while significant agricultural labors are required. This reviewer recommends a re-submission where the authors explore additional potential solutions to poor ART retention other than multi-month dispensing and ART adherence strategies designed for men and/or for younger HIV patients.

The authors recommend further study of the effects of regional differences and climate factors on ART adherence. This would be important when implementing 30-day injectable ART since predicting the beginning of the rainy season would be essential to initiating a 30-day ART injection. The paper referred to the ability of those in Guiana to make predictions as to when the rainy season will begin.

Data collection, cleaning, and analysis details in the methodology section was sufficient to allow others to reproduce this evaluation process with other clinical providers. The authors correctly note that improved tracing systems and clinic transfer procedures would strengthen the number of deaths and self-transfers so that a more accurate assessment of those lost to follow-up could be established. This is also a sound recommendation.

In summary, the paper is of high quality, well organized and clearly written, with only minor edits required. However, the inclusion of the potential of injectable ART as a final recommendation should be made prior to publication so that the paper is most current with the treatment literature.

Reviewer #3: This is a well-written and useful manuscript, thank you.

I have a few minor comments:

1. Although it is mentioned in the limitations, it is not clear in the methods that visits are ascertained at site level only and not across the country. Please explain in the methods that if a patient moves to another facility without documentation, that will be considered LTFU. And whether the unique identifier applies across regions? Line 100- typo: disclosure- disclosure. The description of the setting is otherwise very helpful.

2. Line 112- would it not be more accurate to say “death or transfer not documented in clinic records”

3. Line 142 – typo: on- of

4. Table 1: typo 2020 column 19.8.4

5. Figure5a- has a dashed line that is not explained: please explain

7. PLOS authors have the option to publish the peer review history of their article (what does this mean?). If published, this will include your full peer review and any attached files.

**Do you want your identity to be public for this peer review?** For information about this choice, including consent withdrawal, please see our Privacy Policy.

Reviewer #2: No

Reviewer #3: No

---

## [Decision Letter · Decision Letter 2]

17 Apr 2023

Retention in care among people living with HIV in the national antiretroviral therapy programme in Guinea: a retrospective cohort analysis

PGPH-D-22-01123R2

Dear Vallès,

We are pleased to inform you that your manuscript 'Retention in care among people living with HIV in the national antiretroviral therapy programme in Guinea: a retrospective cohort analysis' has been provisionally accepted for publication in PLOS Global Public Health.

Best regards,

Julia Robinson

Executive Editor

Reviewer Comments (if any, and for reference):

Reviewer's Responses to Questions

**Comments to the Author**

1. If the authors have adequately addressed your comments raised in a previous round of review and you feel that this manuscript is now acceptable for publication, you may indicate that here to bypass the “Comments to the Author” section, enter your conflict of interest statement in the “Confidential to Editor” section, and submit your "Accept" recommendation.

Reviewer #3: All comments have been addressed

2. Does this manuscript meet PLOS Global Public Health’s publication criteria? Is the manuscript technically sound, and do the data support the conclusions? The manuscript must describe methodologically and ethically rigorous research with conclusions that are appropriately drawn based on the data presented.

Reviewer #3: Yes

3. Has the statistical analysis been performed appropriately and rigorously?

Reviewer #3: Yes

4. Have the authors made all data underlying the findings in their manuscript fully available (please refer to the Data Availability Statement at the start of the manuscript PDF file)?

Reviewer #3: Yes

5. Is the manuscript presented in an intelligible fashion and written in standard English?

Reviewer #3: Yes

6. Review Comments to the Author

Reviewer #3: (No Response)

7. PLOS authors have the option to publish the peer review history of their article (what does this mean?). If published, this will include your full peer review and any attached files.

**Do you want your identity to be public for this peer review?** For information about this choice, including consent withdrawal, please see our Privacy Policy.

Reviewer #3: No
